# Comparison of Illumination Methods for Flow-Through Optofluidic Biosensors

**DOI:** 10.3390/mi14040723

**Published:** 2023-03-24

**Authors:** Matthew Hamblin, Joel Wright, Holger Schmidt, Aaron R. Hawkins

**Affiliations:** 1Electrical and Computer Engineering, Brigham Young University, 450 Engineering Building, Provo, UT 84602, USA; 2Electrical and Computer Engineering, University of California, 1156 High Street, Santa Cruz, CA 95064, USA

**Keywords:** optofluidic, lab-on-a-chip, fluorescence, biosensor

## Abstract

Optofluidic biosensors have become an important medical diagnostic tool because they allow for rapid, high-sensitivity testing of small samples compared to standard lab testing. For these devices, the practicality of use in a medical setting depends heavily on both the sensitivity of the device and the ease of alignment of passive chips to a light source. This paper uses a model previously validated by comparison to physical devices to compare alignment, power loss, and signal quality for windowed, laser line, and laser spot methods of top-down illumination.

## 1. Introduction

Optofluidic biosensors and lab-on-a-chip devices have become incredibly useful tools in medical diagnostics. One such device is a sensor that uses fluorescent tag molecules that connect to the DNA in known pathogens, allowing for the detection of bacteria in a matter of hours instead of days, as in typical lab blood testing [1]. For these devices, when a fluorescently labeled probe is attached to a matching DNA sequence, it will fluoresce and produce a detectable light signal when illuminated by a light source. This type of device can be used to detect a variety of medically relevant particles including genetic material from antibiotic resistant bacteria, cancer biomarkers, and viruses such as Ebola, H1N1, and SARS-CoV-2 [1,2,3,4,5,6,7]. The small size and sensitivity of this method of testing allows a relatively small quantity of bacteria or a virus to be detected immediately, as opposed to methods of testing that rely on culturing or amplification.

The model device discussed in this paper consists of a hollow flow-through fluid channel that also guides fluorescent light towards a sensor off the chip [8]. The main methods for exciting a sample in this device are side illumination using a waveguide or illuminating the channel from above. Several factors determine the optimal illumination method [9]. One of these factors is the ease of alignment. In a point of care setting such as a hospital or medical clinic, the need to actively align a laser or optical fiber to the side of the chip is impractical. Packaging a chip with a pre-aligned optical fiber will add costs to the clinical test. Another factor is sensitivity. The collected signal needs to be far enough above the noise threshold and low in variance to produce high-confidence conclusions about the presence of a specific particle. The more sensitive the device, the lower the quantity of the bacteria or virus needed to obtain reliable test results.

One goal for the optofluidic device highlighted here is the use of multiplexing to test for multiple types of bioparticles in one test [1,2]. This is accomplished by using different illumination patterns with different wavelengths of light so that different signals are collected depending on which particle is present in the sample. For instance, if three different bacteria strains are being tested, one fluorescent marker that responds to red light, one that responds to green, and one that responds to blue can be used. If each fluorescent marker connects to a different strain of bacteria and the illumination pattern has a different number of spots (such as seven for red, eight for green, and nine for blue), then, based on the number of events detected, the test can determine which of those strains is present in only one test instead of requiring three separate tests. When multiplexing is used, having a high-intensity and low-variance signal is even more critical because the test needs to not only detect a positive signal but also detect which fluorescent marker has been activated.

The ideal illumination method would have high intensity and low variance, be easy to align, and allow for multiplexing for more rapid testing. Since side illumination through a waveguide is known to require precision alignment, tests have been conducted comparing side illumination to top illumination to determine if top illumination is an acceptable option [10,11]. These tests have shown that top illumination is viable, but the tests conducted to this point have been limited. Several methods are available for illuminating optical biosensors from the top. The methods that will be discussed in this paper are shown in Figure 1.

As shown in Figure 1a, windowed top-down illumination works by shining a relatively large light beam over the top of the channel. The channel is covered in a light blocking layer which has open windows at multiple points along the channel. Using multiple windows allows for some signal-to-noise enhancement though signal processing [11]. This method has the advantage of being easy to align but does not allow for wavelength-based multiplexing. In addition, a significant amount of light power is wasted. The extra light creates the possibility of extra noise in the signal if it is not properly blocked from the channel.

Figure 1b shows that another method for top-down illumination is using a beam splitter or diffractive optical element to create a pattern of multiple illumination areas across the channel [11]. One such method would be to combine a line laser with a diffractive optical element to create a pattern of laser lines across the channel. The line laser method has the advantage of still being easy to align but also allows for wavelength multiplexing. Different patterns could be created for different wavelengths of light, and the extended lines could even allow for multiple channels to be illuminated at once. As is the case with the windowed method, a significant amount of laser power is wasted.

The third method of top-down illumination is to create a laser spot pattern over the channel. This idea is shown in Figure 1c. This method of illumination uses a diffractive optical element combined with a normal spot laser to create a pattern of laser spots. This method has the advantage of little to no wasted laser power and allows for multiplexing but requires precise alignment. A laser spot pattern can be achieved using a DOE such as a beam splitter [12].

Previous simulations have been conducted to compare top-down illumination to side illumination [8]. These simulations have been compared with physical device data to verify their accuracy and have shown that windowed top-down illumination is a viable option for situations where alignment is a concern. Simulated projections have also been conducted to anticipate cases where the fluid sample is hydrodynamically focused, which has been shown to improve the signal quality [13,14,15]. This paper will use the same simulation methods to compare the line laser and spot laser methods to determine which methods provide the best signal sensitivity for both unfocused and 2D hydrodynamic focused cases. Additionally, design parameters such as the laser line and spot size, power efficiency, and alignment sensitivity will be explored to determine when top-down methods make sense for optofluidic devices.

## 2. Biosensor Fabrication

This section includes details on the process of fabricating optofluidic biosensors. These details are included to provide context for the layout, size limitations, and materials used for these devices.

Figure 2 shows an example of how to fabricate the physical flow-through optofluidic biosensor that was used to validate the model data and from which the dimensions of the simulated channel were chosen [8]. First, layers of oxide are deposited over a silicon substrate, resulting in the state shown in Figure 2a. As shown in Figure 2b, a sacrificial layer of a material is deposited in the shape of the fluid channel. As shown in Figure 2c, the oxide layers are etched to create waveguides for signal collection. Figure 2d shows the application of additional oxide layers, followed by the etching of the collection waveguides in Figure 2e. Figure 2f shows the deposition of oxide for cladding layers and the use of acid to etch out the sacrificial photoresist layer, leaving an empty channel for fluid flow.

A microfluidic channel of this type typically has a very small cross-section in the order of 100 square microns. The dimensions for the device considered in this paper are 6 microns of height, 12 microns of width, and 400 microns of length for the fluid channel. The fluid flow in these channels will be laminar. Flow is induced by applying pressure, and the flow profile through the channel can be modeled as parabolic, with particles in the center of the channel flowing faster and particles near the edges of the channel flowing slower. The distribution of the particles within the channel is expected to be random. These biosensors can be further improved with focusing. Focusing pushes the fluid containing the particles being tested towards the center of the channel, which allows for a more consistent flow speed and, therefore, a more consistent signal. For example, two-dimensional focusing can be created by adding two additional channels on either side of the sample channel using the same process of sacrificial photoresist covered in oxide and etched out with acid, as shown in Figure 2 [13]. Pushing liquid through forces the sample into the center where the signal is clearest. An example of this is shown in Figure 3.

## 3. Method

This section describes the process of simulating the different illumination methods described above.

As a fluorescent particle flows through the hollow channel of a flow-through biosensor, it passes through the excitation region, where the channel is illuminated by a laser and emits photons. A fraction of these photons couple into the hollow channel’s mode to a collection waveguide and are transmitted to an off-chip sensor to generate the signal. This process is shown in Figure 4.

As particles flow through the channel, some of the photon signal from the sample will escape the waveguide and be lost.

MATLAB was used to simulate this process of particles flowing through a channel and collecting the generated signal with different types of illumination. A Monte Carlo-type simulation was used to simulate 1000 random particles flowing through the channel. The channel simulated has a width of 12 microns and height of 6 microns.

This simulation works by multiplying an array describing the parabolic fluid flow by an array describing the excitation region. The fluid flow is faster in the center of the channel and slower towards the edges, meaning that particles towards the edge have a longer excitation time. Once the time that a particle spends in the excitation region based on its position is found, this time array is multiplied by the excitation power it travels through to find the total amount of excitation that the particle receives. The excitation region is shown in Figure 5. Figure 5a shows the energy received based on the position without focusing, and Figure 5b shows the energy received based on the position with 2D hydrodynamic focusing. The x position represents the position along the width of the channel, and the y position represents the position along the height of the channel.

Multiplying the array representing the excitation region by the quantum yield of the fluorophore used gives the fluorescent yield of the particle. A sweep of finite differential time domains (FTDT) is then used to simulate the amount of fluorescent energy from a particle that will reach the end of the collection waveguide and be transmitted as a received signal based on the position of the fluorescent particle in the channel. Particles closest to the center of the channel result in a higher fluorescent yield because more photons from them reach the collection site. This method has been compared to test results on real biosensors to validate its accuracy and usefulness [8,11]. The device tested against is a flow-through optofluidic biosensor designed to detect antibiotic-resistant bacteria. It has a 90aM limit of detection [4] and can process samples at a rate of 0.5 µL per minute [2]. This simulation does not consider all factors that can affect the signal quality in real devices, such as differences in biomolecule absorption and fluorescence emissions, variations in laser sources and photodetectors, or signal loss due to reflections. However, the simulation results for top-down illumination using the windowed method were found to closely resemble top-down windowed illumination on a real device when a scaling factor was added to represent loss sources in physical devices that are not calculated as part of the simulation. This shows that the simulation is a valid and relevant tool for comparing illumination methods.

The three different illumination profiles used for the simulations—uniform, one-dimensional gaussian representing a line laser, and two-dimensional gaussian representing a spot laser—are shown in Figure 6a,b,c, respectively.

The simulation method described above was applied to a case that accurately models optofluidic devices which have been fabricated. This simulation considers an optofluidic biosensor illuminated through one window, with one line, or with one spot. This simulation can also be applied to multiplexing, since each additional window, line, or spot will act the same, and the total signal can be found by multiplying by the number of windows, lines, or spots. The simulation conditions are described below. Several design parameters were held constant to provide the most accurate comparison. These parameters are included in Table 1.

The most significant values found from these simulations are the mean and coefficient of variation (CV). The simulation finds the number of events at different intensities measured in units of counts/0.1 ms and then finds the average of these values. The higher the mean, the higher the intensity available for detection by an off-chip detector, as shown in Figure 6. The CV is found by dividing the standard deviation by the mean. The lower the CV, the less variance the signal has and the more predictable the signal is. The best and most usable signal has the highest mean and the lowest CV.

In each case, 1 mW of power was distributed across the excitation region. The differences in the simulation are discussed below for each method.

### 3.1. Windowed Method

For the windowed method, the power was uniformly distributed in a 4-micron-by-12-micron rectangle corresponding to a 4-micron opening across the full width of a 12-micron-wide channel. Simulations were conducted to compare the cases of an unfocused and 2D hydrodynamic focused fluid flow.

### 3.2. Line Laser

To simulate a line laser that had travelled through free space, a one-dimensional gaussian was used, with a full width half max (FWHM) of 4 microns. Simulations were conducted again for the unfocused and 2D focused fluid flows.

### 3.3. Spot Laser

For the spot laser, a two-dimensional gaussian spot was used to simulate a laser spot. Since the spot laser is not uniform across the width of the channel, additional simulations were conducted to compare the effects of spot size. Since perfect alignment is unrealistic in this case, further simulations were conducted to show the effect of misalignment. These simulations were conducted for both unfocused and 2D focused fluid flows.

### 3.4. Power Lost

The total power needed to obtain the desired power in an excitation region can be calculated by taking the ratio of the total illuminated area and the area of the channel that is illuminated. This is shown in Equation (1), where P_total_ is the total power provided by the laser, A_i_ is the illuminated area, Ac is the illuminated area of the channel, P_lost_ is the power lost, P_1_ is the power per excitation region, and n is the number of windows, lines, or spots for a multiplexed case.
P_total_ = n × P_1_ × (A_i_)/(A_c_)(1)

Equation (2) uses the total power found in Equation (1) to calculate the power lost, subtracting the total power from the power incident on the excitation regions.
P_lost_ = P_total_ − n × P_1_(2)

In each case used here, the total length of the channel is considered 400 microns, and the width of the channel is considered 12 microns.

## 4. Results

The results of the simulation described in the previous section are shown in this section. The signal distribution is shown by binning counts/0.1 ms together to show the overall distribution simulated for 1000 particles. The x axis shows the intensity, and the y axis shows the number of events at that intensity.

### 4.1. Windowed Method

The intensity distribution for the windowed method with no focusing is shown in Figure 7. Figure 7 shows a typical example of the random simulation.

As seen in Figure 6, there are more counts of a higher intensity for the focused case than for the unfocused case. This is because when a particle is closer to the center of the channel, a larger portion of the signal it gives off reaches the detector.

Table 2 shows the mean and cv for the windowed case with and without focusing.

As seen in Table 2 above, the 2D focused case has a significantly higher mean and a lower CV. This is because it has been found that particles closest to the center have more of the signal they give off successfully transmitted through the liquid core waveguide to the off-chip sensor.

The windowed method involves using a large laser spot to illuminate the entire area over the channel. To allow for easy alignment, a spot diameter of 600 microns is considered, resulting in a total illuminated area of 2.82 × 105 microns squared. A total of 600 microns was chosen because it allows for 100 microns clearance on either side of the channel. This is the amount of clearance needed to confidently have a chip that snaps into place without precise alignment. The area illuminated over the channel is 4 microns by 12 microns for each of the seven windows, resulting in 336 square microns. This means that, in order to obtain a 7 mW total over the windows, 5.875 W of total power is required. This means that 5.868 W of the power falls outside of the channel, resulting in an illumination efficiency of only 0.1%.

### 4.2. Line Laser Method

The intensity for the unfocused line laser method is shown in Figure 8.

As was the case with the windowed biosensor, 2D focusing causes more of the particles to be in the center of the channel, where the signal given off is more likely to reach the detector.

Table 3 shows the mean and CV for the line laser case with and without 2D hydrodynamic focusing.

Since the total amount of light seen in both cases is 1 mW, the line laser is very similar to the windowed top-down method.

In order to provide easy alignment, 100-micron line lengths were considered. This means that the illuminated area of the channel is 4 by 12 microns for each of the seven lines, or 336 square microns, and the total illuminated area is 4 by 100 microns per line or 2800 square microns total. This means that the total power needed to obtain 1 mW per line is 58.31 mW, resulting in 51.31 mW of power lost, or a 12% illumination efficiency.

### 4.3. Spot Laser

The spot laser case has some significant differences from the windowed and line laser cases. In the other two cases, the illumination pattern did not differ along the 12-micron width of the channel. However, the spot laser has the greatest intensity at the center of the channel. The size of the spot therefore has a significant impact on the signal seen by the channel. Because of this, simulations were conducted that showed the results for the 4-micron case to compare to the dimensions of the previous windowed and line methods, as well as a 20-micron case for a spot size much larger than the width of the channel.

#### 4.3.1. Four Micron Spot Size

The signal distribution for a 4-micron-wide spot laser with and without hydrodynamic focusing is shown in Figure 9.

Similar to the windowed and line laser cases, 2D hydrodynamic focusing increases the signal by causing more of the particles to be in the center, which increases the likelihood of the signal reaching the detector. This is also shown in Table 4.

Table 4 shows that the mean signal is much higher in both the unfocused and focused cases with the 4-micron spot laser than it was for the windowed or line laser method. This is because the light is always focused on the center of the channel. This means that the particles that will result in the greatest likelihood of signal detection also receive the greatest amount of illumination.

#### 4.3.2. Twenty-Micron Spot Size

The signal distribution for a 20-micron-wide spot laser with and without hydrodynamic focusing is shown in Figure 10.

As seen in Figure 10, the signal is improved by hydrodynamic focusing, as with all other cases.

The mean and CV for both the focused and unfocused cases are shown in Table 5.

As seen in Table 5, the average signal much more closely resembles the windowed and line laser cases. This is because as the width of the spot laser increases, the difference in signal intensity along the width of the channel is reduced.

In the case of the spot laser, there is no power lost when the spot is smaller than the width of the channel, and very little power is lost even when the spot becomes larger.

#### 4.3.3. Spot Laser Mean and CV as the Width Changes

Since the width of the laser spot has a significant effect on the amount of power a fluorescent particle sees as it travels through the edges of the channel, a simulation was conducted to find the effect of the width of the laser spot on the mean and CV. The simulation was run 50 times and averaged to reduce the amount of noise from the Monte-Carlo-style simulation. The results of this simulation are shown in Figure 11.

As seen in Figure 11, having a narrow spot size for both the focused and unfocused cases results in a higher mean and lower CV than the line laser and windowed cases. However, as the spot size increases and the variation along the 12-micron width of the channel decreases, the signal becomes much more similar to the windowed and line laser cases.

Additionally, simulations were conducted to find the result of alignment errors. The average signal as a function of alignment errors in microns is shown in Figure 12. This is shown for 4-micron, 6-micron, 10-micron, and 12-micron cases. For these situations, the center of the array with the power in the excitation region was shifted, resulting in less power in the center of the channel and additional power being lost outside of the width of the 12-micron channel. These simulations were run and averaged for 50 tests to reduce the noise in the signal and are shown for both the unfocused and focused cases.

As seen in Figure 12, the smaller the size of the spot, the more dramatically the signal begins to drop off as the alignment shifts away from the center of the channel.

## 5. Comparison and Discussion

The three methods are compared for both the mean counts and CV for the unfocused and focused cases in Table 6. To compare, these are carried out for the 4-micron spot size case.

As shown in Table 6, the uniform and line laser cases are very comparable, the only real difference being the result of randomness in the simulation. The spot laser has a significantly higher average. The CV remains consistent among all methods given the same focusing method.

Table 7 shows the percentage of power wasted for each case, again using the 4-micron case for consistency.

As seen in Table 7, the uniform window method results in almost all incoming laser power being wasted. The line laser method also has significant power lost. For the spot laser, however, no power is wasted if the alignment is within the channel. Less power loss means that money can be saved because a less powerful laser is needed. Additionally, light from the laser that is not in the excitation region has the potential to reflect and introduce noise into the signal.

In systems where the best possible signal is desired, the spot laser is the best solution for illuminating a flow-through biosensor from the top. This method prevents all or most of the power lost and gives the best signal. However, due to the significant change in the signal as the spots are misaligned, this method requires precise alignment that could prove to be neither cost- nor time-effective. This problem is further complicated in situations using multiple spots for multiplexing because that introduces the channel of rotational alignment to the channel. To have a system where a chip can be mechanically clicked into place without the need for proper alignment, a tolerance of around 100 microns is needed. For situations where a detailed alignment is not effective, the windowed and laser line methods allow for much greater alignment tolerance.

The laser line pattern and windowed method have very similar signal intensities, and both have more than enough alignment tolerance. However, the windowed method does not allow for multiplexing.

## 6. Conclusions

Based on the simulations performed and previous device experiment data, using a line laser pattern to illuminate the channel provides the ideal balance of power saved, signal intensity, and ease of alignment. This is beneficial for situations in which a chip is only used once and the ability to quickly align by inserting it into a machine is preferred. In situations where a chip can be used multiple times and the highest signal quality is important enough to invest in a precise alignment process, the spot laser becomes the most beneficial, as it allows for the highest signal quality. Additionally, reducing the amount of power wasted reduces the change by which stray light from other parts of the chip introduces noise.

## Figures and Tables

**Figure 1 micromachines-14-00723-f001:**
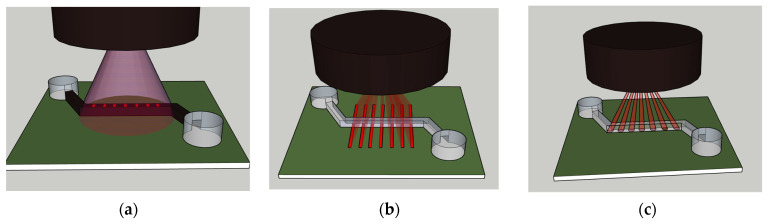
Top-down illumination for a simple optofluidic biosensor using: (**a**) A large laser spot through windows in the fluid channel; (**b**) a pattern of laser lines over the fluid channel; (**c**) a pattern of laser spots.

**Figure 2 micromachines-14-00723-f002:**
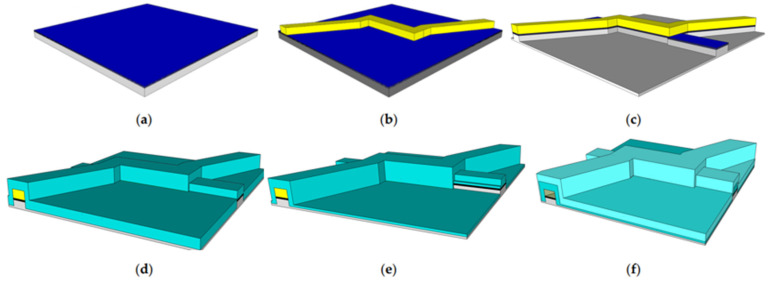
Outline of a fabrication process for a flow-through optofluidic biosensor: (**a**) the substrate; (**b**) sacrificial layer for the hollow channel; (**c**) collection waveguides etched in; (**d**) oxide layer deposition; (**e**) collection waveguides etched; (**f**) cladding layer deposition and sacrificial layer etch.

**Figure 3 micromachines-14-00723-f003:**
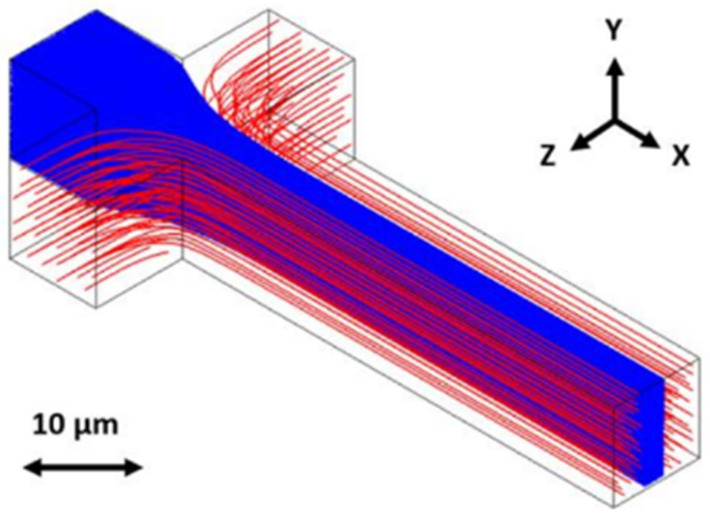
The use of side channels to create 2D focusing. This figure is cited from Hamilton et al. in Micromachines [13] under the Creative Commons Attribution License.

**Figure 4 micromachines-14-00723-f004:**
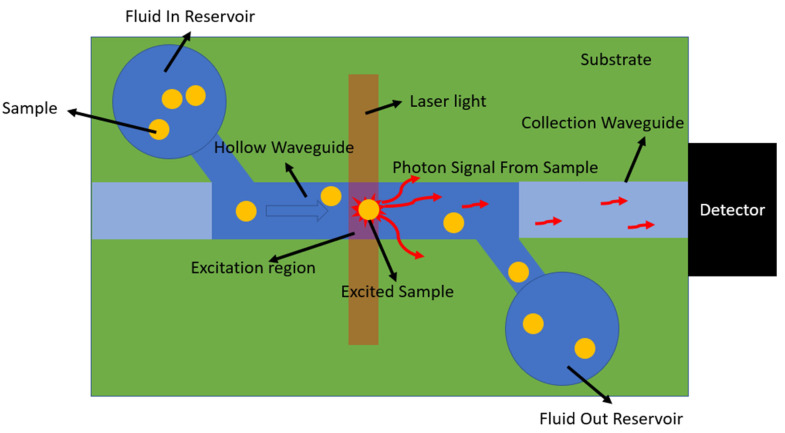
Diagram of a flow-through biosensor showing particles passing through the excitation region and giving off photons.

**Figure 5 micromachines-14-00723-f005:**
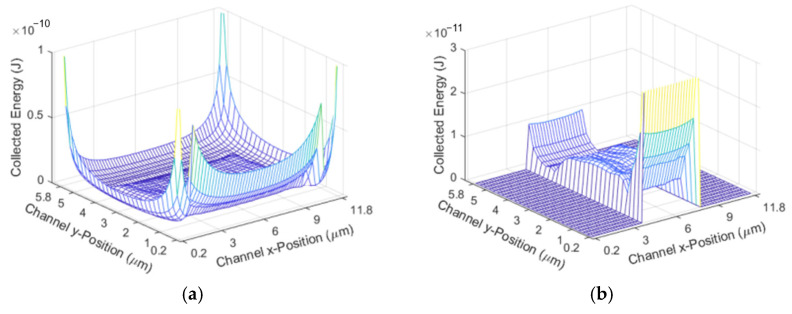
Excitation Region for: (**a**) The unfocused case; (**b**) the 2D focused case.

**Figure 6 micromachines-14-00723-f006:**
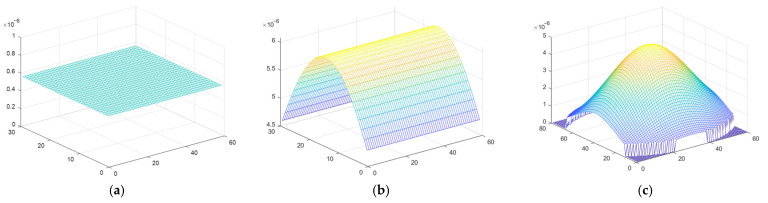
The illumination profile for: (**a**) windowed illumination; (**b**) line laser illumination; (**c**) spot laser illumination.

**Figure 7 micromachines-14-00723-f007:**
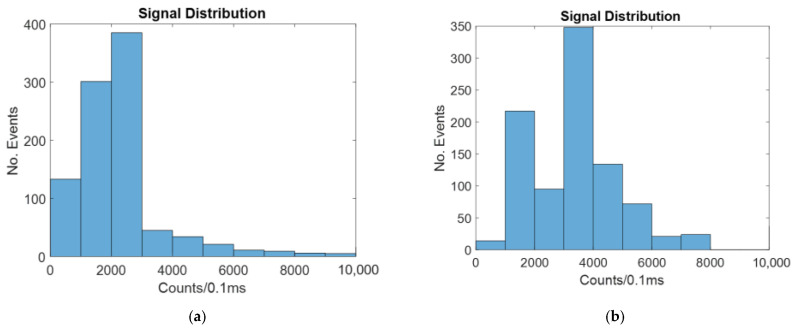
Histograms showing the number of counts at different intensities for a windowed biosensor with: (**a**) no hydrodynamic focusing; (**b**) 2D hydrodynamic focusing.

**Figure 8 micromachines-14-00723-f008:**
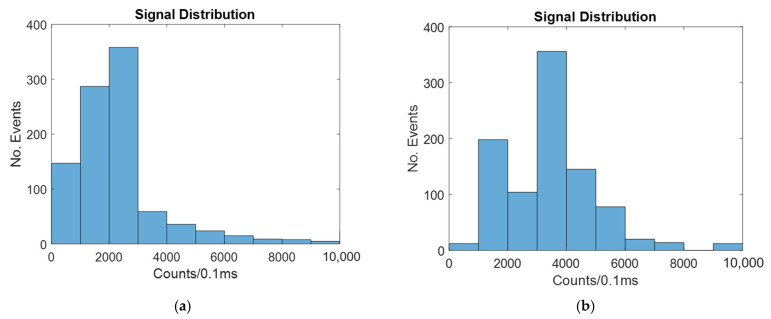
Histograms showing the number of counts at different intensities for a biosensor illuminated by a line laser with: (**a**) no hydrodynamic focusing; (**b**) 2D hydrodynamic focusing.

**Figure 9 micromachines-14-00723-f009:**
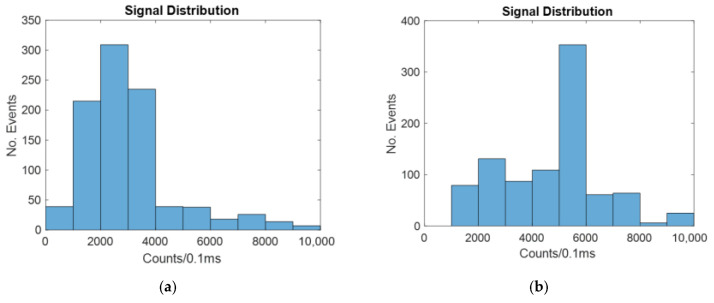
Histograms showing the number of counts at different intensities for a biosensor illuminated by a four-micron spot laser with: (**a**) no hydrodynamic focusing; (**b**) 2D hydrodynamic focusing.

**Figure 10 micromachines-14-00723-f010:**
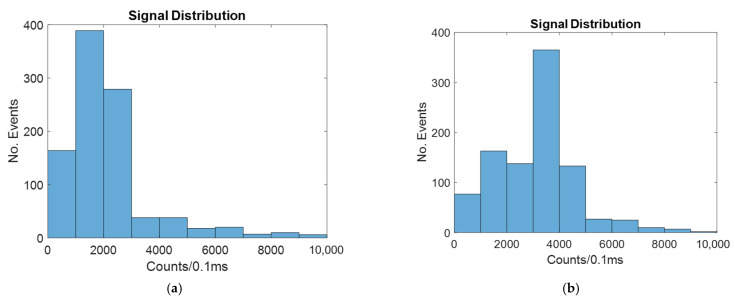
Histograms showing the number of counts at different intensities for a biosensor illuminated by a twenty-micron spot laser with: (**a**) no hydrodynamic focusing; (**b**) 2D hydrodynamic focusing.

**Figure 11 micromachines-14-00723-f011:**
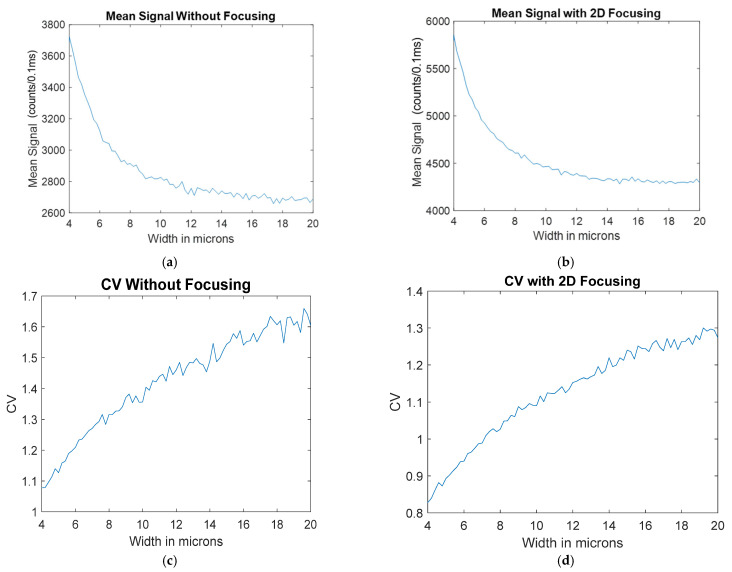
The result of increasing the spot size for a biosensor illuminated with a spot laser including the mean signal as a function of width for: (**a**) no focusing; (**b**) 2D hydrodynamic focusing. The increase in CV as a function of width for: (**c**) no focusing; (**d**) 2D hydrodynamic focusing.

**Figure 12 micromachines-14-00723-f012:**
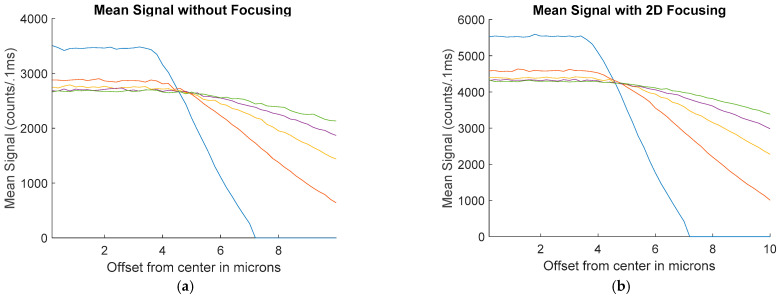
The mean signal as a function of alignment offset for 4-, 6-, 8-, 10-, and 12-micron spot sizes with: (**a**) no focusing; (**b**) 2D hydrodynamic focusing.

**Table 1 micromachines-14-00723-t001:** Design Parameters.

Parameter	Value
Input Power	1 mW
Channel Width	12 microns
Mean Flow Velocity	2 cm/s
Fluorescent Bead Diameter	0.2 microns/s
Excitation Wavelength	633 nm

**Table 2 micromachines-14-00723-t002:** Mean and CV for a windowed biosensor.

Fluid Focusing	Mean Photon Signal (Counts/0.1 ms)	CV
No Focusing	2697	1.09
2D Focusing	4254	0.86

**Table 3 micromachines-14-00723-t003:** Mean and CV for a biosensor illuminated by a line laser.

Fluid Focusing	Mean Photon Signal (Counts/0.1 ms)	CV
No Focusing	2881	1.05
2D Focusing	4296	0.87

**Table 4 micromachines-14-00723-t004:** Mean and CV for a biosensor illuminated by a four-micron-wide spot laser.

Fluid Focusing	Mean Photon Signal (Counts/0.1 ms)	CV
No Focusing	3919	1.09
2D Focusing	6081	0.85

**Table 5 micromachines-14-00723-t005:** Mean and CV for a biosensor illuminated by twenty-micron-wide spot laser.

Fluid Focusing	Mean Photon Signal (Counts/0.1 ms)	CV
No Focusing	2707	1.84
2D Focusing	4416	1.46

**Table 6 micromachines-14-00723-t006:** Comparison of mean and CV for different illumination methods.

Method	No Focusing		2D Focusing	
	Mean	CV	Mean	CV
Uniform	2697	1.09	4254	0.86
Line Laser	2881	1.05	4296	0.87
Spot Laser	3919	1.09	6081	0.85

**Table 7 micromachines-14-00723-t007:** Comparison of power lost for different illumination methods.

Parameter	Value
Uniform Windows	99.9%
Line Laser	88.0%
Spot Laser	as low as 0%, depending on alignment

## Data Availability

The data presented in this study is available upon request from the corresponding author.

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
