# Peer review of "Comparison of Illumination Methods for Flow-Through Optofluidic Biosensors"

_micromachines, 2023, doi:10.3390/mi14040723_

Round 1
Reviewer 1 Report
In this article, the authors have carried out simulations for a top-down illumination for a simple optofluidic biosensor. They have compared power saved, signal intensity, alignment, signal quantity for window method, line laser, and spot laser.
Based on the simulation results presented, line laser pattern to illuminate the channel provides the ideal balance of power saved, signal intensity, and ease of alignment. In situations where a chip can be used multiple times and highest signal quality is important enough to invest in a precise alignment process, the spot laser becomes the most beneficial as it allows for the highest signal quality.
Comments/Suggestions:
1. The authors have carried out detailed simulation for the proposed optofluidic biosensor. The simulation results are very interesting and promising. Only concern is no experimental results presented. In practical implementation, different biomolecules will have different absorptions and fluorescence emissions and the same laser source and photodetector may not work efficiently in terms of signal intensity or signal quality for all the biomolecules in the proposed optofluidic device.
2. Please comment on the limit of detection (LoD) and response time for the proposed optofluidic device.
Reviewer 2 Report
The present manuscript discusses the comparison of different illumination methods for flow-through optofluidic biosensors, which are used as medical diagnostic tools. The article explores the practicality of using optofluidic biosensors in a medical setting, focusing on the sensitivity of the device and the ease of alignment of passive chips to a light source. The manuscript describes simulations that have been done to compare top-down illumination to side illumination and to anticipate cases where the fluid sample is hydrodynamically focused. The paper compares windowed, laser line, and laser spot methods of top-down illumination to determine which method provides the best signal sensitivity for both unfocused and 2D hydrodynamic-focused cases. Overall, this is a well-written article aiming to provide insights into how different illumination methods can affect the sensitivity and practicality of optofluidic biosensors in a medical setting. My suggestion is to accept the article as is.
Author Response
No comments to respond to.